# Results of a Cluster Randomized Controlled Trial to Promote the Use of Respiratory Protective Equipment among Migrant Workers Exposed to Organic Solvents in Small and Medium-Sized Enterprises

**DOI:** 10.3390/ijerph16173187

**Published:** 2019-08-31

**Authors:** Wen Chen, Tongyang Li, Guanyang Zou, Andre M.N. Renzaho, Xudong Li, Leiyu Shi, Li Ling

**Affiliations:** 1Department of Medical Statistics, School of Public Health, Sun Yat-sen University, Guangzhou 510080, China; 2Center for Migrant Health Policy, Sun Yat-sen University, Guangzhou 510080, China; 3Institute for International Health and Development, Queen Margaret University, Edinburgh EH21 6UU, UK; 4School of Social Science and Psychology, Western Sydney University, Penrith 2751, Australia; 5Guangdong Prevention and Treatment Center for Occupational Diseases, Guangzhou 510300, China; 6Department of Health Policy and Management, Bloomberg School of Public Health, Johns Hopkins University, Baltimore, MD 21205, USA

**Keywords:** respiratory protective equipment, migrants, workers, small and medium-sized enterprises, cluster randomized controlled trial

## Abstract

*Background*: Existing evidence shows an urgent need to improve respiratory protective equipment (RPE) use, and more so among migrant workers in small and medium-sized enterprises (SMEs). The study aimed to assess the effectiveness of a behavioral intervention in promoting the appropriate use of RPE among internal migrant workers (IMWs) exposed to organic solvents in SMEs. *Methods*: A cluster randomized controlled trial was conducted among 1211 IMWs from 60 SMEs in Baiyun district in Guangzhou, China. SMEs were deemed eligible if organic solvents were constantly used in the production process and provided workers with RPE. There were 60 SMEs randomized to three interventions on a 1:1:1 ratio, namely a top-down intervention (TDI), a comprehensive intervention, and a control group which did not receive any intervention. IMWs in the comprehensive intervention received a module encompassing three intervention activities: An occupational health education and training component (lectures and leaflets/posters), an mHealth component in the form of messages illustrative pictures and short videos, and a peer education component. The TDI incorporated two intervention activities, namely the mHealth and occupational health education and training components. The primary outcome was the self-reported appropriate RPE use among IMWs, defined as using an appropriate RPE against organic solvents at all times during the last week before measurement. Secondary outcomes included IMWs’ occupational health knowledge, attitude towards RPE use, and participation in occupational health check-ups. Data were collected and assessed at baseline, and three and six months of the intervention. Generalized linear mixed models were performed to evaluate the effectiveness of the trial. *Results*: Between 3 August 2015 and 29 January 2016, 20 SMEs with 368 IMWs, 20 SMEs with 390 IMWs, and 20 SMEs with 453 IMWs were assigned to the comprehensive intervention, the TDI, and the control group, respectively. At three months, there were no significant differences in the primary and secondary outcomes among the three groups. At six months, IMWs in both intervention groups were more likely to appropriately use RPE than the control group (comprehensive intervention: Adjusted odds ratio: 2.99, 95% CI: 1.75–5.10, *p* < 0.001; TDI: 1.91, 95% CI: 1.17–3.11, and *p* = 0.009). Additionally, compared with the control group, the comprehensive intervention also improved all three secondary outcomes. *Conclusions*: Both comprehensive and top-down interventions were effective in promoting the appropriate use of RPE among IMWs in SMEs. The comprehensive intervention also enhanced IMWs’ occupational health knowledge, attitude, and practice. *Trial registration*: ChiCTR-IOR-15006929. Registered on 15 August 2015.

## 1. Background

Along with China’s rapid economic development and urbanization over the past three decades, the number of internal migrant workers (IMWs) has continued to grow rapidly and reached 282 million in 2016 [1]. Unlike international migrant workers who cross borders, IMWs do not leave their country of birth. However, under the household registration management system in China, IMWs are classified as being temporary residents in host cities because it is difficult for them to change their household registration regardless of where they live and work. Furthermore, the majority of IMWs in China are rural-to-urban migrants who are less educated and have poor health literacy [2]. Consequently, they are predominantly employed in certain kinds of labor to perform menial blue-collar jobs, internationally known as the 3D (dirty, dangerous, and difficult or demeaning) jobs [3]. Available evidence in China [4,5,6] and abroad [7,8,9] consistently shows that both internal and international migrant workers are younger, less educated, work for longer hours and in the worse occupational environments (e.g., higher risks of occupational hazards exposures), and have poor occupational health outcomes than native workers. Among IMWs, those working in the small and medium-sized enterprises (SMEs), where occupational health services have limited reach, are the most vulnerable population [10]. 

In China, enterprises shall, as required by laws and regulations, take measures to ensure that employees receive occupational safety and health services [11,12]. In addition to protecting and promoting workers’ health through setting up occupational disease hazards assessment and monitoring systems, adopting technologies, processes, equipment, and materials to protect workers’ occupational health in the process of production, and organizing regular occupational health training and check-ups, enterprises should also provide individual workers with personal protective equipment (PPE) and associated training, maintenance, and supervision. PPE use is a defense measure against exposure to occupational disease hazards. However, there are different types of PPE. For example, to prevent health risks from exposure to organic solvents, such as acute and chronic occupational poisoning and occupational allergic disease, respiratory protective equipment (RPE) and gloves should be appropriately used all the time during the work [13,14].

Our previous work in China showed that, in Guangdong province, only 16.8% to 22.4% of IMWs in SMEs reported using PPE at all times during work, including RPE and ear protector [15,16]. This low uptake of risk reduction practices observed in China has also been reported among workers in SMEs in Nigeria [17] as well as international migrant workers in the USA [18] and Australia [19]. In the most recent systematic review, Luong and colleagues found that existing behavioral interventions to promote workers’ use of RPE had methodological limitations, grading the quality of evidence as very low with small sample sizes ranging from 44 to 206. Additionally, the review found that there was no intervention targeting IMWs in SMEs, which remains a critically under-researched topic [20]. 

Within this context, the research team has developed the first SME-based behavioral intervention to promote the use of RPE among IMWs [21]. The intervention design was based on relevant behavioral change theories, including the Health Belief Model [22], Bandura’s Social Cognitive Theory [23], and the Theory of Reasoned Action [24], and the empirical evidence regarding individual and organization factors impacting IMWs’ utilization of RPE [15,16,25]. Guided by these behavioral change theories, the intervention encompassed behavioral change activities that would improve IMWs’ awareness of the health effects associated with the exposure to organic solvents and their knowledge of why, when, where, and how to use RPE, and increase IMWs’ self-efficacy in the appropriate use of RPE. In addition, literature reviews suggested interpersonal and organization factors should be considered in the intervention design, such as role models, subjective norms, supportive interpersonal relationships, and workplace environment.

Findings from two systematic reviews of occupational health interventions found that passive health education and training may not generate changes in the use of RPE among workers in SMEs [20,26], and concluded that migrant and SME-sensitive interventions are urgently needed. The adoption of mHealth [27] which broadly encompasses the use of mobile communication devices such as mobile phones, tablets, and wearable devices to deliver health care services and information, and peer education [28] have been successfully used to increase or promote the uptake of activities geared towards preventing a wide range of health problems. Therefore, these health promotion activities might also be relevant in work-based interventions targeting IMWs to overcome the low uptake of RPE [29]. The low uptake of RPE among migrants is often driven by cultural factors, customs, and language barriers [2]. Consequently, we have designed a multi-pronged behavioral intervention by combining health education and training, mHealth, and peer education components [21]. Therefore, the aim of this study was to use a cluster randomized controlled trial (CRCT) design to assess the effectiveness of a multi-pronged behavioral intervention in promoting the appropriate use of RPE among IMWs exposed to organic solvents in SMEs. We hypothesized that a comprehensive (three components: Occupational health education and training, mHealth, and peer education) and top-down (two components: Occupational health education and training and mHealth) interventions would be more effective than a control condition (no intervention) in promoting the appropriate use of RPE among IMWs at three- and six-month assessments.

## 2. Methods

### 2.1. Trial Design

Between 3 August 2015, and 31 January 2016, we conducted a three-arm CRCT designed to investigate the effectiveness of SME-based multipronged behavioral interventions for the appropriate use of RPE among IMWs. Internationally, the definitions of SMEs vary country by country, but the most used definition conceptualizes SMEs according to the number of workers employed by the enterprise, its level of capital investment or assets, and its volume of production or business turnover [30]. In China, an SME refers to an enterprise employing 20–1000 workers, with an annual turnover of 3–400 million Chinese Yuan [31]. The Chinese definition was retained for this study. In addition, in this study, SMEs were considered as “clusters”, but we measured all outcomes at the participants’ level with a structured self-reported questionnaire at baseline (before the implementation of interventions); and at three and six months of the intervention. This trial has been registered with the Chinese Clinical Trial Registry (ChiCTR-IOR-15006929). The study protocol has been described previously [21]. This trial has been approved by the Institute Review Board of School of Public Health, Sun Yat-sen University (Ref. 02/ 2014). All IMWs signed written informed consent to participate in the trial.

### 2.2. Settings and Participants

This study was conducted among IMWs exposed to organic solvents from 60 eligible SMEs in Baiyun district, Guangzhou city, China. SMEs were deemed eligible for this trial if organic solvents were constantly used in their production processes and provided workers with occupational RPE. IMWs were defined as Chinese workers with a non-registered permanent residence in Guangzhou city. Guangzhou remains a major destination for IMWs in China, and Baiyun district is the most industrially developed area in the city. By the end of 2014, there were 1.39 million internal migrants in the Baiyun district, accounting for 60.8% of the total population [32]. In this study, the scope of occupational hazards was limited to organic solvents because they rank as one of the top occupational hazards in China [13] and workers exposed to different organic solvents are required to use the same type of RPE, namely organic respirator, thus allowing for comparison across enterprises.

### 2.3. Randomization

According to the information provided by the local administration of work safety, which is the supervision and administration department for occupational safety and health, a total of 861 SMEs were eligible for participation in the study. To get a representative sample of eligible SMEs in the eligible study sites, a simple random sampling approach was used to select 60 SMEs from the 861 SMEs (60 SMEs were estimated to be a sufficient sample size, see later). Specifically, all the eligible SMEs were listed in ascending order by enterprise name and were assigned a computer-generated random number from one to 861. Then SMEs with the random number from one to 60 were selected. If a selected SME refused to participate, we approached the SME with the next random number until a total of 60 SMEs were recruited. All selected SMEs received the same general information in Mandarin which outlined the purpose and procedures of this study through their managers, and the managers were given one week to decide whether or not to participate in this study. In total, 66 SMEs were contacted, and 60 agreed to participate, giving a response rate of 90.9%. Then, the eligible 60 SMEs were randomly assigned to either the intervention or the control arm on a 1:1:1 ratio. Given that the size of SMEs (small or medium) was not related to the use of RPE among IMWs [16], the randomization of SMEs to either the intervention or control arm did not take into account the size of the enterprise. The randomization and intervention allocation were performed by a trial statistician who was blind to the study.

Within each selected SME, only IMWs were approached to participate because they are more vulnerable to occupational hazards exposures and are less likely to adopt personal protective behaviors than native workers [4,5,6,15,16]. With help from occupational health personnel in the included SMEs, all eligible IMWs were recruited if they were first-line production workers exposed to organic solvents, were on duty on the day of the baseline survey, had been employed for more than one month prior to the survey, and agreed to participate in the trial. First-line production team leaders and workers unable to read or communicate were excluded. All eligible IMWs in the same SME received the same intervention (see the flowchart in Figure 1).

### 2.4. Intervention Procedures

Two intervention groups were compared with a control group. All intervention activities were implemented during a six-month period after the baseline assessment and were delivered in Mandarin, which is the official language in China, and the language used at work. All participants could speak Mandarin and they did not report language barriers. In the control group, we did not provide any intervention. Details of interventions procedures are summarized in Table 1.

The first intervention group implemented a top-down intervention (TDI). The TDI encompassed two components: Occupational health education and training module and mHealth (Table 1). The occupational health education and training module included one lecture on occupational health for managers and occupational health personnel and passive occupational health education and training activities for IMWs. The occupational health education for managers and occupational health personnel was delivered by the principal investigator to help them recognize the enterprises’ responsibilities and obligations towards their IMWs’ occupational health, the benefits of healthy workplaces, and good occupational health practices. Trained trainers (research team members) delivered the passive health education/training sub-component. The training included three sessions (1 h each) of the standard training provided by the Guangdong Prevention and Treatment Center for Occupational Diseases. Trainees also received one lecture covering the demonstration of the appropriate use of RPE and fit testing/user-seal check, complemented with two brochures, and two posters to IMWs. The second’s component of the TDI was the mHealth, which encompassed messages, illustrative pictures, and short videos sent to IMWs twice per week via instant message apps. All materials could be read/watched from either mobile phones or computers. To increase viewing, we embedded links of related contents in each text. App hits were recorded to monitor viewing. Passive health education/training and mHealth aimed to improve IMWs’ awareness of the consequences of organic solvents exposure and benefits of the appropriate and effective use of RPE.

The second intervention was a comprehensive intervention that incorporated the TDI and a peer education component. According to the number of recruited IMWs in each SME, one or more peer groups with a group size of eight to 15 people were organized. Though self-nomination, one member of the group was assigned as a group leader (peer educator) and received two sessions (1 h each) of standard training on peer education, complemented with a 33-page instruction manual that group leaders could take home. The instruction manual introduced themes, main contents, and flow charts of each peer education session (45–60 min/session). Additionally, during the 6-month period, trained trainers provided weekly online assists to all peer educators. Then, peer educators were asked to organize a total of six peer education sessions over the study period (one every month) and monitor group members’ utilization of RPE during work. To monitor the intervention delivery, peer educators were required to fill in an implementation form and take photos of each session. In return, each peer educator received 50 RMB (8 $USD) as a token of appreciation. Moreover, trained trainers evaluated peer educators’ performances based on submitted records and photos, including compliance with the study protocol, the number of participants in each peer education session, and progress compared with the last monitoring month. Every month, the performance of all peer educators was evaluated, and the top five best performing educators received an award in the form of an additional 50 RMB. In addition to the purpose of the TDI, by providing peer education, the comprehensive intervention also aimed to help IMWs identify and reduce barriers, and create role models and supportive interpersonal relationships and safe workplace environments.

The process evaluation was undertaken at various stages of the project to monitor the intervention’s implementation fidelity, compliance with the protocol, dropout rate, and the quality and consistency in intervention activities. The implementation of the intervention across SMEs was monitored by trained trainers and peer educators using a standard implementation form and taking photos of intervention participation. The intervention implementation-related data were sent to the steering group every month to ascertain the extent of implementation and quality. Additionally, IMWs’ self-reported adherence was collected by questionnaires. Participants’ adherence to the intervention, such as the number of peer education sessions attended, was calculated based on trained trainers, peer educators, and IMWs’ feedback.

### 2.5. Outcome Measures

All outcome measures were collected at baseline (before any intervention), and at three and six months of the intervention. During measurements, SME managers and owners were requested to be absent to avoid their impacts on the participants’ responses. The primary outcome was the self-reported appropriate use of RPE during the last week prior to the survey (yes/no). IMWs reported whether they had used RPE against organic solvents (yes/no), the type of RPE used (disposable mask, half mask, and full-face mask), and the frequency of RPE utilization (always, most of the time, sometimes, rarely, or never) during the last week. According to the national guideline on the use of RPE against hazards from organic solvents’ exposure [14], IMWs who always use an appropriate type of organic respirator during the last week can be categorized into the group identified as appropriately using RPE. Two data administrators, who did not deliver the interventions, determined whether participants used RPE appropriately or not based on process data.

Secondary outcomes included occupational health knowledge, attitude towards RPE utilization, and participation in occupational health check-up during the past six months. The occupational health knowledge scale contained ten questions (yes/no) related to risks of exposure to organic solvents and related hazards, and how to choose and use RPE (Cronbach’s alpha = 0.78). Questions were developed based on the Chinese national standards for the use of RPE against potential health hazards from organic solvents [14]. The score of occupational health knowledge ranged from zero to 10, with a higher score indicating better knowledge. Attitude towards RPE utilization was measured by a five-point Likert instrument developed and validated by the researchers [25]. The instrument included four factors (Cronbach’s alpha = 0.84), i.e., willing of use (two items), self-efficacy (three items), perceived benefits (two items), and perceived barriers (two items). Attitudes towards RPE utilization was generated by summing scores of the nine items. The score ranged from nine to 45, with a higher score indicating a positive attitude. Questions for measuring secondary outcomes are listed in Appendix A.

### 2.6. Statistical Analysis

Sixty SMEs with 920 IMWs were estimated to be a sufficient sample size to detect a 43% and 53% increment in the proportion of IMWs appropriately using RPE in the TDI and comprehensive intervention groups respectively, compared with the proportion in control group (16.8%). A power of 80%, a significance level of 0.05, intra-class correlation coefficient (ICC) of 0.1, and attrition of 20% among migrants and 10% among SMEs (due to SMEs bankruptcy) were used in sample size estimation [21].

Analyses were done at the participant level. Following the intention-to-treat (ITT) principle [33], all IMWs that were initially allocated to interventions and provided data regarding the appropriate use of RPE at baseline were included in the final analysis. All analyses were done using IBM SPSS Statistics 21.0 (IBM Corp., Armonk, NY, USA). Mean, standard deviation (SD), frequency, and proportion of IMWs’ characteristics (age, sex, education level, marital status, duration of migration, employment duration in the current position, weekly working hours, social models for RPE use [25], and interpersonal support for RPE use [25]) and baseline data of primary and secondary outcomes were calculated for each group and tested with a model controlled for clustering of IMWs within SMEs. The ICC was calculated for the primary outcome to measure variability across SMEs (clusters).

To test differences in the primary outcome and participation in occupational health check-ups among three intervention groups, we used generalized linear mixed models (GLMMs) with a logistic link, a random effect to account for clustering of IMWs within SMEs, and a nested random effect to account for repeated measures within IMWs. For occupational health knowledge and attitude towards RPE utilization, GLMMs with a normal link, a random effect for clustering, and a nested random effect for repeated measures were used. The GLMMs for each outcome included fixed effects for the intervention group, the measurement point (baseline, three months, and six months), a group-by-month interaction; controlling for individual demographic characteristics. Based on our previous work on factors impacting IMWs’ RPE utilization [15,16,25], individual demographic characteristics controlled for in the models included age, sex, education, marital status, duration of migration, duration of current position, weekly working hours, social models for RPE use, and interpersonal support for RPE use. To test the significance of the effectiveness, we calculated the unadjusted odds ratio (uOR) or regression coefficients (*b*) and adjusted odds ratios (aOR) or regression coefficients (*b_ad_*) and their 95% confidence intervals (CIs) for each of the two intervention groups compared with the control group at three and six months.

A subgroup analysis was performed to test the effect of IMWs’ adherence to peer education. The comprehensive intervention group was divided into two subgroups: Attended one to three peer education sessions versus attended four to six sessions. The effectiveness of two subgroups on the primary outcome was compared with the control group. To test modifying effects of age, sex, and education on the effectiveness of intervention, we performed we included an age-by-group interaction, a sex-by-group interaction, and an education-by-group interaction in the GLMMs for the primary outcome. The modifying effect of SMEs’ occupational health practice was not tested because all study SMEs provided free PPE and performed supervision on RPE utilization (Appendix A).

## 3. Results

Figure 1 shows a flowchart of this study. There were 20 SMEs with 373 IMWs, 20 SMEs with 393 IMWs, and 20 SMEs with 459 IMWs were assigned to the comprehensive intervention, the TDI, and the control group, respectively. Of the 1211 IMWs with baseline assessment (368 in the comprehensive intervention, 390 in the TDI, and 453 in the control group), 973 (comprehensive intervention: 276 (75%); TDI: 340 (87%); and control: 357 (79%)) and 903 (comprehensive intervention: 247 (67%); TDI: 324 (83%); and control: 332 (73%)) participated in the survey at three months and six months, respectively. The current study had an attrition rate of 25.4% at six months, which was significantly lower than the 88.5% attrition rate reported in another enterprise-based intervention among IMWs at six months in China [34].

Table 2 shows baseline characteristics of internal migrant workers in each group. The mean age of 1211 IMWs was 34.8 years (SD = 10.0), male workers accounted for 71.9%, and 31.2% of participants had high school and above degrees. The average duration of exposure to organic solvents in the current position was 63.6 months (SD = 56.6). There was no significant difference among the three groups after controlling for clustering of IMWs within SMEs. Baseline characteristics of the 60 study SMEs were listed inAppendix A. The size of study SMEs was between 20 and 400 workers. All the SMEs provided free PPE and occupational safety and health training to employees and performed regular supervision on workers’ use of PPE. However, six out of 60 SMEs did not organize occupational health check-ups.

Overall, at the end of the six months intervention, the proportion of IMWs appropriately using RPE increased by 20.2 percentage points (54.6% to 74.8%) in the comprehensive intervention group and by 12.0 percentage points (50.5% to 62.5%) in the TDI group, while the control group remained at almost the same level of 52% over the period of the study (Table 3). The ICC for the appropriate use of RPE across SMEs was 0.428 at baseline, 0.439 at three months, and 0.440 at six months.

Table 3 shows the effectiveness of the interventions on the primary outcome. At three months of the intervention, both of the unadjusted and adjusted interaction between groups and measurement points show no significant effect on the appropriate use of RPE in the two intervention groups, compared with the control group. However, at six months, IMWs in both intervention groups reported a higher likelihood of appropriately using RPE than the control group (comprehensive intervention: aOR = 2.99, 95% CI: 1.75–5.10, *p* < 0.001; TDI: aOR = 1.91; 95% CI: 1.17–3.11; and *p* = 0.009). We further analyzed the effectiveness of two subgroups in the comprehensive intervention (attended one to three sessions of peer education [*n* = 271, 73.6%] versus four to six sessions [*n* = 97, 26.4%]). There were significant effects of the subgroup with good peer education attendance than the control group (aOR = 2.97, 95% CI: 1.37–6.43, and *p* = 0.006 at three months, aOR = 13.72, 95% CI: 4.81–39.14, and *p* < 0.001 at six months). However, no significant effects of the subgroup with poor peer education attendance were found compared with the control group.

None of the migrants’ age, sex, and education significantly modified the intervention effect for the appropriate use of RPE (interaction *p*-value = 0.852 for age-by-group, 0.851 for sex-by-group, and 0.396 for education-by-group). Details of these analyses are, therefore, not shown in this article.

The interactions between groups and time of measurement showed that, at three months, there were no significant effects on all three secondary outcomes of both intervention groups compared with the control group (Table 4). However, at six months, we noted significant effects on all secondary outcomes of the comprehensive intervention compared with the control group (*b_ad_* = 0.32, 95% CI: 0.01–0.61, and *p* = 0.043 for occupational health knowledge; *b_ad_* = 1.82, 95% CI: 0.67–2.96, and *p* = 0.002 for attitude towards RPE utilization; aOR = 2.13, 95% CI: 1.20–3.81, and *p* = 0.010 for participation in occupational health check-up). No adverse events were observed during the intervention.

## 4. Discussion

This enterprise-based CRCT sought to the effectiveness of a multipronged behavioral change intervention in promoting the appropriate use of RPE among IMWs. The findings suggest that passive health education/training combined with mHealth (TDI) or TDI and peer education (comprehensive intervention) were significantly more effective in promoting the appropriate use of RPE than the control group (no intervention). Furthermore, IMWs’ peer education attendance was positively associated with the effects of the intervention. To date, there are no behavioral interventions targeting production workers reporting significant effects compared with no intervention [20,35,36]. Our findings that the comprehensive intervention resulted in improving RPE utilization among IMWs, occupational health knowledge, attitude towards RPE utilization, and participation in occupational health check-ups are novel and a great addition to the literature. The significant effects of our interventions could be mainly due to the robustness of the study design. With IMWs and SMEs’ involvements, this tailored and theory-driven intervention program was developed based on existing and new evidence related to improving RPE use among IMWs.

First, our intervention was behavioral change theory-driven and covered individual, interpersonal, and organization factors related to the use of RPE among IMWs in SMEs [21]. According to the Health Belief Model [22], people will take a health action if they feel it will be effective in addressing a negative health issue and they believe they are able to identify and implement an appropriate course of action. Guided by this theory, we designed passive occupational health education and training and mHealth to help IMWs recognize the negative health issue (i.e., health consequences of exposure to organic solvents) and the appropriate use of RPE against hazards from exposure to organic solvents. Under the social cognitive theory [23], the human functioning is viewed as the product of a dynamic interplay of personal, behavioral, environmental influences, and self-efficacy beliefs stand at the very core of this theory. Previous research has also shown that self-efficacy is a critical mediator for both behavioral initiation and maintenance [37] Informed by this theory, the research team and trained peer educators provided behavior trainings and health education to increase IMWs’ self-efficacy in the appropriate use of RPE and peer education was designed to help the participants establish supportive interpersonal relationships and workplace environments. The theory of reasoned action [24] assumes that people will have strong intentions to take action if they have positive attitudes towards the behavior and if they believe that important others think they should perform it (i.e., having subjective norm). Therefore, we delivered intensive health education to build the participants’ positive attitudes towards the use of RPE. Moreover, peer education also aimed to develop role models and subjective norms.

Second, the intervention adopted an interactive education/training approach, which is more effective than passive education/training in improving workers‘ occupational health behavior and knowledge [38,39]. Consequently, compared with the majority of interventions in which workers passively received education/training [20,35,40,41], we incorporated an interactive peer education component. An interactive peer education approach has been found to enhance social modelling and interpersonal support, which were positively related to migrant workers’ compliance with RPE use [25] and were important for behavior change maintenance [42].

Third, instead of providing a few sessions of education/training at the initial stage in most of the existing interventions [40,41,43], we developed interventions over a six-month period. Education materials were delivered twice a week via mHealth to produce sustained efforts [42]. Our findings suggest that the impact of intervention at three months was not significant in both intervention groups, suggesting that adopting a new occupational behavior program may require a long-term process. This observation is consistent with the available evidence, which suggests that it takes on average 91 days to form a healthy behavior in everyday life and model habit in the real world [44]. Future study should ensure adequate duration of the occupational intervention to generate changes in behavior.

Only the comprehensive intervention showed significant effects on improving IMWs’ occupational health knowledge, attitude towards RPE utilization, and participation in occupational health check-ups. Our findings confirm the advantage of peer education in occupationally behavioral intervention among IMWs. They also indicate that changes in attitude and knowledge are more likely to occur if workers receive interventions from their peers than experts. According to Kelman’s theory of attitude change [45], IMWs in the TDI group may follow the “compliance” process, which means they adopted the induced behavior only because they expected to get a favorable reaction from researchers or SMEs managers. In contrast, the participants who received interventions from their peers with similar values might have induced behavior via “identification” or “internalization.” That is, IMWs use RPE because they consider this behavior to be related to their desired relationships with peers. Therefore, from the perspective of behavioral maintenance, the comprehensive intervention that includes both TDI and peer education is a better choice for SME-based behavioral intervention programs than the TDI. Furthermore, compared with the control group, significant effects were only found among individuals who attended more than half of peer education sessions. On the one hand, the result emphasized the importance of adequate duration of the intervention; on the other hand, it indicated participants’ compliance was crucial to the effectiveness of the comprehensive intervention. Compliance is a key factor that impacts the effectiveness of health interventions and services and, correspondingly, is often set as a target of interventions [46,47]. In this study, participants with good peer education attendance only accounted for 26.4% of the comprehensive intervention group’s sample size. Strategies to improve IMWs’ compliance with behavioral interventions are topics worthy of further study.

Noteworthily, although the majority of study SMEs provided occupational health services in accordance with the relevant laws and regulations [11,12], IMWs’ baseline occupational health knowledge, attitude, and behavior remained unsatisfactory. Compared with IMWs who received regular occupational health training provided by the SMEs (i.e., the control group), those in the comprehensive intervention group had significant better occupational health behavior in general. But this benefit was not found in the TDI group. The findings indicate that peer education programs might be integrated into regular occupational health training to improve the SMEs practice on occupational health in general, not only for PPE.

Limitations of this study include measuring outcomes by self-report data, as with other similar studies [35,36,39]. Although recall bias and reporting bias are inevitable, collecting self-report data is much more feasible than continuous observation in workplace settings, and research showed self-reported PPE use was highly consistent with field observation [48]. Furthermore, the sample used in this study was drawn from one metropolitan and SMEs in China, so the generalizability of findings may be limited. However, to improve the diversity of the study samples, we chose Baiyun district, a typical and well-established industrial area, as the research site, and the study SMEs involved various industries, including furniture, leather goods, paints and coatings, plastic and plastic cement, and electronic manufacturing (Appendix A). Attrition was another limitation, while attrition in migrants and workplace studies are not unique [34,49]. In the current study, the attrition rate was much lower than previous intervention among internal migrant workers in China (25.4% versus 88.5 % at six months) [34]. Attrition in this study appeared mainly in the third month (October), which is the seasonal harvesting peak and also a returning peak for internal migrants. In total, out of 238 IMWs did not participate in the second round of the survey, 114 (47.9%) were because of absenting from the SME.

## 5. Conclusions

This CRCT provides much-needed empirical evidence of the effectiveness of behavioral interventions in SMEs by showing that passive health education/training combined with mHealth and peer education can promote the appropriate use of RPE and related occupational health knowledge and attitude, and participation in occupational health check-ups among IMWs. In China and other low and middle-income countries where coverage of basic occupational health services remains low, interactive strategies could be integrated into regular occupational health services to enrich service forms and enhance service effect. In order, to develop work-based occupational intervention programs, more effort will be needed to improve participants’ attendance.

## Figures and Tables

**Figure 1 ijerph-16-03187-f001:**
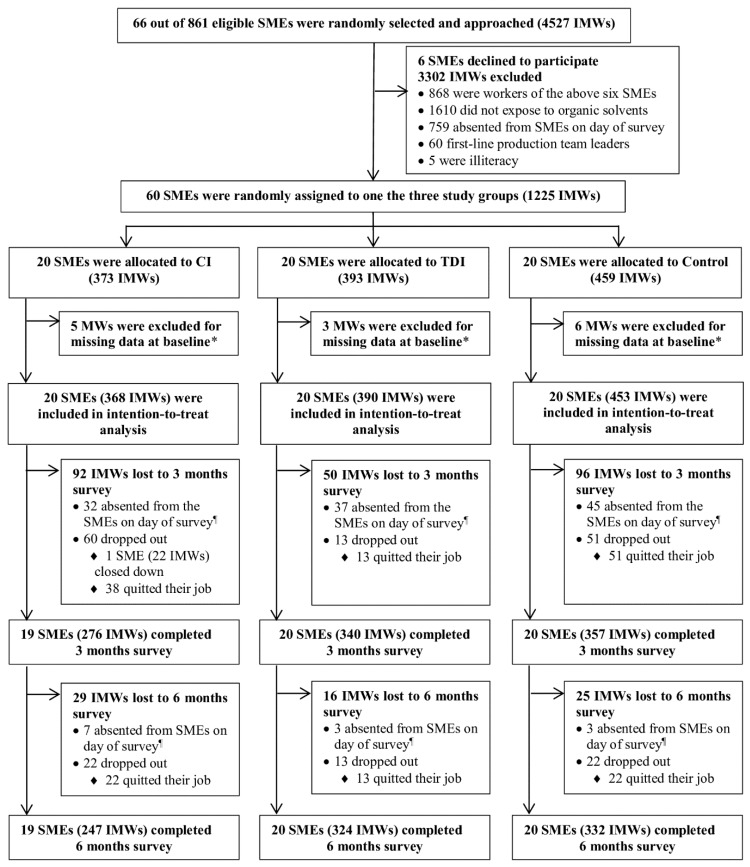
**Flow diagram.** SMEs = small and medium-sized enterprises; IMWs = internal migrant workers; CI=comprehensive intervention; TDI =top-down intervention. * Internal Migrant workers who had missing data for the primary outcome (effective use of personal protective equipment during the last week) at baseline survey were excluded. ^¶^ Internal Migrant workers absented from the SME on day of survey because of work shifts or days off.

**Table 1 ijerph-16-03187-t001:** Intervention procedures during a six-month period.

Intervention	Component	Target Participants	Content	Frequency	Delivered by
Top-down intervention	Occupational health education	Managers and occupational health personnel in SMEs	enterprises’ responsibilities on occupational health benefits of healthy workplaces good occupational health practices	One lecture (1 h) The 1st week of the intervention	The principal investigator of the study
	Passive occupational health education and training—lectures	Internal migrant workers	introduction of organic solvents risks of organic solvents exposure benefits of RPE, when and where to use how to use and maintain RPE demonstration of RPE wearing and fit testing/user-seal check	One lecture (1 h) The 1st week of the intervention	Trained trainers (3 sessions (1hr each) of training provided by Guangdong Prevention and Treatment Center for Occupational Diseases)
	Passive occupational health education and training—brochures and posters	Internal migrant workers	risks of organic solvents exposure benefits of RPE, when and where to use illustrative pictures of RPE wearing and fit testing/user-seal check	Brochures delivered twice (baseline and three months), posters posted in the workspace	Trained trainers
	mHealth	Internal migrant workers	risks of organic solvents exposure when, where and how to choose, use, testing, and maintain RPE real stories on organic solvents poisoning laws and regulations on occupational health protection puzzles and answers	Twice per week	Trained trainers via Instant Message Apps
Comprehensive intervention	Top-down intervention	As same as the top-down intervention			
	Peer education	Internal migrant workers	ice-breaking game and introduction of peer education organic solvents and occupational protection training on RPE use and fit testing/user-seal check, share personal experiences personal experiences of benefits and barriers to use RPE, how to reduce barriers how to maintain RPE utilization	Monthly (45–60 min/session)	Trained peer educators (2 sessions (1hr each) of face-to-face training and weekly online assists provided by trained trainers)

SMEs = small and medium-sized enterprises; RPE = respiratory protective equipment.

**Table 2 ijerph-16-03187-t002:** Baseline characteristics of 1211 study participants in intervention and the control groups.

Characteristics	Comprehensive Intervention (20 SMEs, *n* = 368)	Top-down Intervention (20 SMEs, *n* = 390)	Control (20 SMEs, *n* = 453)	Total (60 SMEs, *n* = 1211)	*p* *
Age (years), mean (SD)	33.5 (10.8)	36.3 (9.5)	34.4 (9.7)	34.8 (10.0)	0.386
Sex, *n* (%)					0.532
Male	245 (66.6)	291 (74.6)	335 (74.0)	871 (71.9)	
Female	123 (33.4)	99 (25.4)	118 (26.0)	340 (28.1)	
Education, *n* (%)					0.762
Primary School	50 (13.6)	43 (11.0)	39 (8.6)	132 (10.9)	
Secondary School	197 (53.5)	236 (61.5)	268 (59.2)	701 (57.9)	
High School and above	121 (32.9)	111 (28.5)	146 (32.2)	378 (31.2)	
Marital status, *n* (%)					0.102
Married	239 (64.9)	310 (79.5)	322 (71.1)	871 (71.9)	
Single	129 (35.1)	80 (20.5)	131 (28.9)	340 (28.1)	
Duration of migration (years), mean (SD)	10.5 (7.3)	12.4 (7.0)	11.6 (6.8)	11.5 (7.0)	0.143
Employment duration in the current position (months), mean (SD)	59.0 (56.7)	67.1 (50.2)	64.6 (61.2)	63.6 (56.6)	0.765
Weekly working hours, mean (SD)	55.2 (8.7)	55.1 (9.0)	55.6 (9.7)	55.3 (9.2)	0.920
Social models for RPE use † (score range: 2–10), mean (SD)	7.9 (2.5)	7.8 (2.4)	7.8 (2.6)	7.8 (2.5)	0.813
Interpersonal support for RPE use† (score range:3–15), mean (SD)	9.8 (3.7)	10.0 (3.3)	10.0 (3.6)	9.9 (3.5)	0.945

SD = standard deviation; SMEs = small and medium-sized enterprises; RPE = respiratory protective equipment. * *p*-value of models controlled for clustering of IMWs within SMEs. † Social models for RPE use include two questions and Interpersonal support for RPE use includes three questions [25]. The score range of social models for RPE use and interpersonal support for RPE use is 2–10 and 3–15, respectively, with higher scores indicating more positive models or stronger support.

**Table 3 ijerph-16-03187-t003:** Intervention effectiveness on the appropriate use of respiratory protective equipment (RPE) (primary outcome) at 3- and 6-month of intervention (*n* = 1211).

Group	Baseline		3-Month Follow-up						6-Month Follow-up					
	*n*	Case (%)	*n*	Case (%)	Our (95%CI)	*p*	aOR (95%CI)	*p*	*n*	Case (%)	Our (95%CI)	*p*	aOR (95%CI)	*p*
CIG(all)	368	201 (54.6)	276	170 (61.6)	1.31 (0.84,2.05)	0.227	1.33 (0.81,2.19)	0.261	246	184 (74.8)	**2.80 (1.73,4.52)**	**<0.001**	**2.99 (1.75,5.10)**	**<0.001**
CIG(≤3 sessions)	271	148 (54.6)	179	98 (54.7)	0.94 (0.57,1.55)	0.800	0.86 (0.49,1.51)	0.594	168	113 (67.3)	**1.82 (1.08,3.07)**	**0.024**	1.74 (0.96,3.16)	0.070
CIG(≥4 sessions)	97	53 (54.6)	97	72 (74.2)	**2.53 (1.26,5.05)**	**0.009**	**2.97 (1.37,6.43)**	**0.006**	78	71 (91.0)	**10.73 (4.21,27.36)**	**<0.001**	**13.72 (4.81,39.14)**	**<0.001**
TDI	390	197 (50.5)	339	207 (61.1)	1.52 (0.99,2.33)	0.053	1.55 (0.96,2.50)	0.072	323	202 (62.5)	**1.80 (1.16,2.78)**	**0.008**	**1.91 (1.17,3.11)**	**0.009**
Control	453	235 (51.9)	357	191 (53.5)	Reference	-	Reference	-	332	170 (51.2)	Reference	-	Reference	-

CIG(all) = All participants in the comprehensive intervention group; CIG(≤3 sessions) = participants in the comprehensive intervention group who attended 1–3 sessions of peer education; CIG(≥4 sessions) = participants in the comprehensive intervention group who attended 4–6 sessions of peer education; TDI = top-down intervention group. RPE = respiratory protective equipment. our = unadjusted odds ratio; aOR = adjusted odds ratio; CI = confidence interval. ORs and 95% CIs were generated from generalized linear mixed models with a logistic link, and aORs were generated by adjusting for age, sex, education, marital status, duration of migration, duration of current position, weekly working hours, social models for RPE use, and interpersonal support for RPE use. Bold figures: Significant differences.

**Table 4 ijerph-16-03187-t004:** Intervention effectiveness on secondary outcomes at 3- and 6-month of intervention (*n* = 1211).

Outcomes	Group	Baseline		3-Month Follow-up						6-Month Follow-up					
		*n*	Mean(SD)	*n*	Mean(SD)	*b* (95%CI)	*p*	*b_ad_* (95%CI)	*p*	*n*	Mean(SD)	*b* (95%CI)	*p*	*b_ad_* (95%CI)	*p*
Occupational health knowledge	CIG	368	6.6 (1.9)	241	7.2 (1.9)	0.26 (−0.14,0.67)	0.203	0.15 (−0.25,0.54)	0.464	211	7.6 (1.8)	**0.46 (0.04,0.85)**	**0.032**	**0.32 (0.01,0.61)**	**0.043**
	TDI	390	6.9 (2.0)	270	7.4 (1.7)	0.20 (−0.20,0.59)	0.332	0.21 (−0.17,0.60)	0.249	297	7.3 (1.8)	−0.11 (−0.49,0.27)	0.570	−0.09 (−0.46,0.29)	0.651
	CG	453	6.6 (2.0)	293	6.9 (2.0)	Reference	-	Reference	-	299	7.3 (1.7)	Reference	-	Reference	-
Attitude towards RPE utilization	CIG	368	36.3 (5.8)	241	38.3 (4.9)	**1.17 (0.07,2.26)**	**0.037**	0.69 (−0.35,1.74)	0.194	211	39.4 (5.0)	**2.25 (1.07,3.43)**	**<0.001**	**1.82 (0.67,2.96)**	**0.002**
	TDI	390	37.0 (5.3)	270	38.2 (5.0)	0.63 (−0.43,1.70)	0.242	0.30 (−0.73,1.32)	0.571	297	38.5 (5.6)	1.03 (−0.07,2.12)	0.066	0.92 (-0.15,1.98)	0.091
	CG	453	37.1 (6.2)	293	37.9 (5.2)	Reference	-	Reference	-	299	37.9 (5.3)	Reference	-	Reference	-
		***n***	**Case (%)**	***n***	**Case (%)**	**Our (95%CI)**	***p***	**aOR (95%CI)**	***p***	***n***	**Case (%)**	**Our (95%CI)**	***p***	**aOR (95%CI)**	***p***
Participation in occupational health check-up	CIG	365	189 (51.8)	235	173 (74.3)	**1.65 (1.00,2.71)**	**0.050**	1.68 (0.98,2.89)	0.059	206	163 (79.2)	**2.18 (1.28,3.70)**	**0.004**	**2.13 (1.20,3.81)**	**0.010**
	TDI	390	235 (60.3)	269	206 (76.4)	1.38 (0.84,2.28)	0.206	1.38 (0.79,2.38)	0.256	291	215 (73.6)	1.06 (0.65,1.74)	0.814	1.03 (0.63,1.70)	0.901
	CG	451	252 (55.9)	291	192 (66.3)	Reference	-	Reference	-	299	205 (68.3)	Reference	-	Reference	-

CIG = comprehensive intervention group; TDI = top-down intervention group; CG = control group. SD = standard deviation; *b* = unadjusted linear regression coefficients; *b_ad_ =* adjusted linear regression coefficients; our = unadjusted odds ratio; aOR = adjusted odds ratio; CI = confidence interval. ORs and 95% CIs were generated from generalized linear mixed models with a logistic link, *b*s and 95% CIs were generated from generalized linear mixed models with a normal link; and aORs and *b_ad_ s* were generated by adjusting for age, sex, education, marital status, duration of migration, duration of current position, weekly working hours, social models for RPE use, and interpersonal support for RPE use. Bold figures: Significant differences.

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
