# Peer review of "Results of a Cluster Randomized Controlled Trial to Promote the Use of Respiratory Protective Equipment among Migrant Workers Exposed to Organic Solvents in Small and Medium-Sized Enterprises"

_ijerph, 2019, doi:10.3390/ijerph16173187_

Round 1

Reviewer 1 Report

    This is an interesting and relevant study. The experimental design seems appropriate. One of the major limitations is the self-report data which was addressed by authors in the discussion.

    The results are presented in table format, which works but can be overwhelming for the readers. The manuscript can benefit from graphs that complement the table and highlighting the significant differences using asterisk. 

Author Response

Point 1: This is an interesting and relevant study. The experimental design seems appropriate. One of the major limitations is the self-report data which was addressed by authors in the discussion.

The results are presented in table format, which works but can be overwhelming for the readers. The manuscript can benefit from graphs that complement the table and highlighting the significant differences using asterisk. 

Response 1: Thank you for your positive comments. We prefer table format because it gives details of the estimated effect size (odds ratio/ linear regression coefficient) and its 95% confidence interval. However, we highlighted the significant differences in bold to make the tables more clear for readers.

Reviewer 2 Report

About Background:

This research present the case of internal migrants workers but the introduction refers to real migrant workers.

I do not feel comfortable mixing both collectives. Please identify, if any, the reference of the situation of internal migrants and if any similarities and differences between both types of migrants.

At the same time references to SME are refering to countries where like in Europe SME´s have less than 250 workers. Please present the distribution of the number of workers in the SME participating.

Methods:

Please note that you have published the protocol somewhere else. Fig 2 need to be citing the original source and pieces of information in the article too.

Nevertheless, authors need to indicate the questions delivered for measuring the secondary outcome and the descriptive statistics of each of the results.

Regarding secondary outcome, it seems authors are measuring a composite indicator, or the evaluation of a latent variable. Please clarify and communicate how this variable is obtained for the regression model use.

Discussion and conclusions

Please do not discuss migrant workers. A worker without a permanent residence can be a national or a foreign worker. It is not appropriate to attribute these results to the concept of migrant workers. Note that a migrant from other country can have a permanent residence. Possibly more than migrants we are dealing with low-income workers from countryside around. Migrants are related not only with low income but with different culture, language and safety attitudes. Please try to identify some information of the workers participating as nationality.

Please highlight the importance of the number of sessions in the outcomes.

Minor notes: Please check letter size as sometimes there are words in a different letter size. In Table 3 and 4 there is a typo as uOR is used instead of aOR sometimes and others b is used instead. In Table 4 it seems that a title line is repeated.

Author Response

Point 1: About Background: This research present the case of internal migrants workers but the introduction refers to real migrant workers. I do not feel comfortable mixing both collectives. Please identify, if any, the reference of the situation of internal migrants and if any similarities and differences between both types of migrants.

Response 1: We thank the reviewer for his/her suggestion. As we explained in the background section, internal migrant workers in China are very simliar with international migrant workers, in terms of the work environment and occupational health and safety. We have thoroughly revised the background section to include further details about the similarities and differences between both types of migrants, and identify references related to internal migrant workers in China. In addition, we have now consistently used internal migrant workers through out the text instead of migrant workers to distinguish the two groups of migrant workers.

Point 2: At the same time references to SME are refering to countries where like in Europe SME´s have less than 250 workers. Please present the distribution of the number of workers in the SME participating.

Response 2: Characteristics, including the distribution of the number of workers, of the study SMEs were listed in Appendix Table S2. We have also added the description in the results section, where we elaborated the appendix.

Point 3: Methods: Please note that you have published the protocol somewhere else. Fig 2 need to be citing the original source and pieces of information in the article too.

Response 3: The study protocol has been published on the BMC Public Health and has been cited as reference #21.

We guess the reviewer meant Fig 1 because there is only one figure in this manuscript. Figure 1 is the flowchart of this study.

Point 4: Nevertheless, authors need to indicate the questions delivered for measuring the secondary outcome and the descriptive statistics of each of the results.

Regarding secondary outcome, it seems authors are measuring a composite indicator, or the evaluation of a latent variable. Please clarify and communicate how this variable is obtained for the regression model use.

Response 4: We thank the reviewer for his/her suggestion. We have now provided questions delivered for measuring secondary outcomes as Appendix Table S1. Descriptive statistics of each of the results were listed in Table 3 and Table 4 (Baseline column).

There are three secondary outcomes in this study. Occupational health knowledge is the sum of the score of the ten questions (ranging from zero to ten), which were developed and validated by the researchers. Attitudes towards respiratory protective equipment (RPE) utilisation was generated by summing scores of nine items developed and validated by the researchers. The score of attitude towards RPE utilisation ranged from nine to 45. The third outcome, participation in occupational health check-up during the past six months was measured by a dichotomous variable (yes/no). For the first two outcomes, their scores were used as dependent variables in GLMMs with a normal link. To test differences in general occupational health behaviour among three groups, we used GLMMs with a logistic link.

Point 5: Discussion and conclusions

Please do not discuss migrant workers. A worker without a permanent residence can be a national or a foreign worker. It is not appropriate to attribute these results to the concept of migrant workers. Note that a migrant from other country can have a permanent residence. Possibly more than migrants we are dealing with low-income workers from countryside around. Migrants are related not only with low income but with different culture, language and safety attitudes. Please try to identify some information of the workers participating as nationality.

Response 5: We thank the reviewer for this suggestion. First of all, as we described in inclusion criteria of participants, all migrant workers in this study are Chinese workers with a non-registered permanent residence in Guangzhou city. However, we did not collect nationalities of the study participants. Although China is a multinational country, the Han population accounts for around 92% of the total population in China.1

Secondly, we think the discussion on effectiveness of this intervention can not be out of context of migrant workers. First, this tailored intervention program was developed based on evidence related to improving RPE use among internal migrant workers. For example, we designed mhealth and peer education to overcome RPE use-related difficulties among migrants, such as trust, culture, custom, and language. Moreover, as we introduced in the background section, internal migrant workers in China are similar with international migrant workers in terms of work environment and occupational health and safety.

1: Population Census Office under the State Council, Department of Population and Employment Statistics National Bureau of Statistics. Tabulation on the 2010 population census of the People’s Republic of China. China Statistics Press. Beijing. 2011. http://www.stats.gov.cn/english/statisticaldata/censusdata/rkpc2010/indexch.htm

Point 6: Please highlight the importance of the number of sessions in the outcomes.

Response 6: We appreciate the reviewer’s suggestion. We have now added discussion on the importance of the number of sessions in the outcomes.

Point 7: Minor notes: Please check letter size as sometimes there are words in a different letter size. In Table 3 and 4 there is a typo as uOR is used instead of aOR sometimes and others b is used instead. In Table 4 it seems that a title line is repeated.

Response 7: We have carefully checked and revised letter size.

In table 3 and table 4, uOR is unadjusted odds ratio, aOR is adjusted odds ratio, b is unadjusted linear regression coefficients, and bad is adjusted linear regression coefficients. aORs and bad were generated by models adjusted for age, sex, education, marital status, duration of migration, duration of the current position, weekly working hours, social models for RPE use, and interpersonal support for RPE use. We have explained these abbreviations in the statistical analysis section and table note of table 3 and table 4. In Table 4, two title lines were used to distinguish uOR/aOR and b/bad.

Reviewer 3 Report

Abstract

P1 lines 33-34: is it possible to include the analytical model such as generalized linear mixed models (GLMMs)? I believe it will give good indication to the readers. P1 line 39: include “95% CI:” after TDI: 1.91 P1 lines 43-44: use ‘coma’ instead of ‘and’ after occupational health knowledge….. Moreover, I would suggest to remove ‘general occupational health’ in line 44 and keep only ‘behavior’ if you did not explain properly what is ‘general occupational health behavior’. In P7 lines 222-223, it is vaguely defined.

Background

P2 lines 61-65: please rewrite and make readable P2 line 64: explain ‘PPE’ P2 lines 76-77: if you really want to mention the name of the theories, you should explain them briefly. P2 line 88: mHealth-it would be great to have a brief introduction on ‘mHealth’ Overall, the knowledge gap and significance of the study are poorly constructed. For example, statements related to organic solvents and their health consequences along with the usefulness of RPE usages among SME workers could strengthen the significance of the study.

Materials and method

P3 line 111: I think it should be ….“self-reported”…. P7 lines 222-223: general occupational health behavior outcome is vaguely defined. Authors mentioned in the background that they have health belief model then why do not you call knowledge, attitude, and practice because ‘behavior’ is the combination of all components. P7 line 248: I would prefer ‘general occupational health practice’ instead of “general occupational health behavior” throughout the manuscript. P7, Statistical analysis: name and the selection of confounders are not clear. P7-8 lines 263-264: Authors stated that they tested effect modification, but they did not mention anything about it in the abstract. I would request to include sentences related to effect modification result in the abstract. The dropout was about 50% in the second round. Did it affect the result? If not, how do you explain it? If yes, how did you tackle it?

Results

P9 line Table 2: Was there any other category of marital status despite-single and married? If so, how did you categorize them into only two?

Discussion

Overall, the discussion needs improvement. Discuss the reason of wider Confidence Intervals (CIs) such as P9 lines 303-309: “(comprehensive intervention: aOR= 2.99, 95% CI: 1.75-5.10, p<0.001; TDI: aOR=1.91, 95% CI: 1.17-3.11, p=0.009).” and “(aOR=2.97, 95% CI: 1.37-6.43, p=0.006 at three months, aOR=13.72, 95% CI: 4.81-39.14, p<0.001 at six months).” Discuss the null result of effect modification and plausible reasons. More discussion in the line of theories related to health behavior. Did you notice any influence of SME managers or owners to manipulate the attitude and practice of workers despite the interventions induced practices? If so, how do you explain it along with the effectiveness of interventions?

Conclusion

Any policy recommendations for national and global occupational health and safety stakeholders.

Author Response

Point 1: Abstract

P1 lines 33-34: is it possible to include the analytical model such as generalized linear mixed models (GLMMs)? I believe it will give good indication to the readers. P1 line 39: include “95% CI:” after TDI: 1.91 P1 lines 43-44: use ‘coma’ instead of ‘and’ after occupational health knowledge….. Moreover, I would suggest to remove ‘general occupational health’ in line 44 and keep only ‘behavior’ if you did not explain properly what is ‘general occupational health behavior’. In P7 lines 222-223, it is vaguely defined.

Response 1: We thank the reviewer for his/her suggestions. We have now revised the abstract correspondingly.

Point 2: Background

P2 lines 61-65: please rewrite and make readable P2 line 64: explain ‘PPE’ P2 lines 76-77: if you really want to mention the name of the theories, you should explain them briefly. P2 line 88: mHealth-it would be great to have a brief introduction on ‘mHealth’ Overall, the knowledge gap and significance of the study are poorly constructed. For example, statements related to organic solvents and their health consequences along with the usefulness of RPE usages among SME workers could strengthen the significance of the study.

Response 2: We appreciate the reviewer for his/her thoughtful suggestions. We have rewritten the background section and added introduction on mHealth and statements related to health consequences of organic solvents exposure and the protective effect of RPE. Line 104-109 on page 3 explained the behavioural change theories in the context of this study. In addition, we have added more explanations and applications of the behavioural change theories in the discussion section.

Point 3: Materials and method

P3 line 111: I think it should be ….“self-reported”…. P7 lines 222-223: general occupational health behavior outcome is vaguely defined. Authors mentioned in the background that they have health belief model then why do not you call knowledge, attitude, and practice because ‘behavior’ is the combination of all components. P7 line 248: I would prefer ‘general occupational health practice’ instead of “general occupational health behavior” throughout the manuscript. P7, Statistical analysis: name and the selection of confounders are not clear. P7-8 lines 263-264: Authors stated that they tested effect modification, but they did not mention anything about it in the abstract. I would request to include sentences related to effect modification result in the abstract. The dropout was about 50% in the second round. Did it affect the result? If not, how do you explain it? If yes, how did you tackle it?

Response 3: We have defined the “general occupational health behavior” as participation in occupational health check-up during the past six months. Confounders and the selection basis were provided in the Statistical analysis section.

As we explained in the results section, none of the migrants' age, sex, and education significantly modified the intervention effect for the appropriate use of RPE. In addition, effect modification is not a recommended item which needs to be reported in the abstract based on the Consolidated Standards of Reporting Trials for abstracts. Therefore, we did not report effect modification results in the abstract.

The dropout was 10.2% (124/1211) in the second round of survey instead of 50%. The reasons for dropout were the participants quitted their job (n=102), and one SME closed down (n=22). No dropout was due to the participants did not want to report their RPE utilisation status. Therefore, the dropout individuals can be considered as ignorable missing data.1 In this case, generalized linear mixed models can eliminate complete-case bias by incorporating all available information, which means the dropout does not affect the result.

1: Ibrahim JG, Molenberghs G. Missing data methods in longitudinal studies: a review. Test. 2019; 18(1):1-43.

Point 4: Results

P9 line Table 2: Was there any other category of marital status despite-single and married? If so, how did you categorize them into only two?

Response 4: In the questionnaire, there are four categories of marital status, namely single, married, divorced, and widowed. However, there were only 11 divorced and four widowed participants. Therefore, we categorized them into single individuals.

Point 5: Discussion

Overall, the discussion needs improvement. Discuss the reason of wider Confidence Intervals (CIs) such as P9 lines 303-309: “(comprehensive intervention: aOR= 2.99, 95% CI: 1.75-5.10, p<0.001; TDI: aOR=1.91, 95% CI: 1.17-3.11, p=0.009).” and “(aOR=2.97, 95% CI: 1.37-6.43, p=0.006 at three months, aOR=13.72, 95% CI: 4.81-39.14, p<0.001 at six months).” Discuss the null result of effect modification and plausible reasons. More discussion in the line of theories related to health behavior. Did you notice any influence of SME managers or owners to manipulate the attitude and practice of workers despite the interventions induced practices? If so, how do you explain it along with the effectiveness of interventions?

Response 5: For intervention effectiveness on dichotomous outcomes, e.g., RPE utlilisation, the width of the confidence interval is determined by the degree of confidence, sample size, and the number of positive events.2 Therefore, in this study, the wider 95%CIs for aORs in the subgroup analysis than the 95%CIs for aORs based on the overall sample could be due to a smaller number of participants and a smaller number of participants who appropriately used RPE in subgroups than the overall sample.

  In this study, we did not find differing effects of age, sex, and education on the RPE utlilisation, depending on the intervention group. Therefore, the modification effect was not statistically significant. Because existing literature did not suggest that there might be significant interactions between these demographic factors and occupational practice among migrant workers who received different types of occupational health education, we did not include discussion on the null result of effect modification.

We have now added more discussion on the application of behavioural change theories in this study.

For the last question, during the intervention process, we did not receive any feedback from the participants about SME managers or owners tried to manipulate or manipulated their attitude and practice. In fact, we tried to avoid this situation by requiring SME managers or owners to be absent during measurements. This information has been added to the outcome measures section.

2. Brożek J, Falavigna M. Chapter 12.3: What Determines the Width of the Confidence Interval? In Users' Guides to the Medical Literature: A Manual for Evidence-Based Clinical Practice, 3rd ed. McGraw-Hill Education. New York. 2014

Point 6: Conclusion

Any policy recommendations for national and global occupational health and safety stakeholders. 

Response 6: We have now added brief policy recommendations.